# Dynamic tracking of objects in the macaque dorsomedial frontal cortex

Rishi Rajalingham [1,2,7], Hansem Sohn [3,4,7] & Mehrdad Jazayeri [1,5,6] ✉

A central tenet of cognitive neuroscience is that humans build an internal model of the external world and use mental simulation of the model to perform physical inferences. Decades of human experiments have shown that behaviors in many physical reasoning tasks are consistent with predictions from the mental simulation theory. However, evidence for the defining feature of mental simulation – that neural population dynamics reflect simulations of physical states in the environment – is limited. We test the mental simulation hypothesis by combining a naturalistic ball-interception task, large-scale electrophysiology in non-human primates, and recurrent neural network modeling. We find that neurons in the monkeys' dorsomedial frontal cortex (DMFC) represent task-relevant information about the ball position in a multiplexed fashion. At a population level, the activity pattern in DMFC comprises a low-dimensional neural embedding that tracks the ball both when it is visible and invisible, serving as a neural substrate for mental simulation. A systematic comparison of different classes of task-optimized RNN models with the DMFC data provides further evidence supporting the mental simulation hypothesis. Our findings provide evidence that neural dynamics in the frontal cortex are consistent with internal simulation of external states in the environment.

With just a few glances, humans can make rich inferences about objects and predict their future states. For example, we can predict how balls would move around a billiard table from random initial configurations[1] or the probability that a perturbation would tear down a tower of blocks[2]. A dominant cognitive theory posits that humans make these predictions by forming mental models of the physical world and using those models to perform simulations[3–5]. In support of this theory, it has been shown that human predictions are consistent with noisy stochastic simulations of abstract models instantiated by high-level computer programs[2,6]. Despite the success of the simulation-based theory ('intuitive physics engine'), alternative model-free approaches have also been proposed to account for human physical reasoning behaviors[7]. Indeed, humans can readily identify heuristic solutions for tasks without building complex mental models and

running simulations postulated by the former theory. Recent experiments supported this alternative by showing that humans rely on an efficient, good-enough, but rather inflexible sensorimotor mapping to perform physical inference[8,9]. Therefore, the debate remains as to the degree to which humans rely on mental simulations to make judgments about physical scenes.

Central to the debate is whether the brain recursively updates the internal state of the objects over short time intervals using the mental model of physical scenes[10,11]. A large body of human psychophysical experiments sought to tackle this question by testing behavioral predictions from the alternative theories[1,2,8,9,12,13]. However, these approaches are inherently limited as behavioral readouts are typically low-dimensional and, therefore, cannot reveal the rich and dynamic nature of the mental simulations. On one hand, neuroimaging studies in

¹McGovern Institute for Brain Research, Massachusetts Institute of Technology, Cambridge, MA, USA. ²Reality Labs, Meta; 390 9th Ave, New York, NY, USA. ³Center for Neuroscience Imaging Research, Institute for Basic Science (IBS), Suwon, Republic of Korea. ⁴Department of Biomedical Engineering, Sung-kyunkwan University (SKKU), Suwon, Republic of Korea. ⁵Department of Brain & Cognitive Sciences, Massachusetts Institute of Technology, Cambridge, Massachusetts, USA. ⁶Howard Hughes Medical Institute, Massachusetts Institute of Technology, Cambridge, USA. ⁷These authors contributed equally: Rishi Rajalingham, Hansem Sohn. ✉e-mail: mjaz@mit.edu

humans have attributed elevated activity in various brain areas to mental simulations. However, human neuroimaging methods do not have the resolution needed to verify that the underlying brain dynamics are indeed consistent with such simulations[14–17]. Therefore, evidence for the defining feature of mental simulation – that the brain reproduces neural states that correspond to physical states – is lacking[7,18]. On the other hand, neuroscientists have used electrophysiological recording in animals that were trained to perform object-tracking tasks that required latent object representations beyond simple sensorimotor mapping. For example, non-human primates were trained to make choices based on stimulus properties of invisible objects, and successful performance of these tasks suggested that the animals might learn the objects' permanence and movement. These studies further found that single neurons in the frontal eye field[19–22], posterior parietal cortex[23,24], medial superior temporal area[25], and ventral premotor cortex[26] are sensitive to latent task variables like directions or speeds of the invisible motion stimuli. These sensitivities of single neurons can serve as a basis for mental object tracking, yet it remains unclear whether neural population activity can be directly linked to the moment-by-moment state of physical objects.

Building on those previous works, we aimed to test the mental simulation hypothesis by analyzing neural population dynamics during object tracking in the macaque frontal cortex. When designing our experiment, we considered two important desiderata. First, we wanted animals to be able to learn fast and generalize flexibly. To that end, we designed an intuitive task that capitalizes on the monkey's understanding of object permanence. In this task, which we refer to as 'mental pong,' animals control a virtual paddle with a joystick to intercept a moving object, which is initially visible but then moves behind a two-dimensional occluder. We randomly varied the properties of the moving ball across trials, including its initial position, speed, directions, and interaction with the boundaries of the 2-dimensional arena. These variations promote flexible mental simulation strategy and discourage animals from solving the task using fixed input-output mappings, which is a viable solution when sensorimotor contingencies have fewer variations[21,24,26] (but see[23]). We further allowed animals to freely move their eyes and hands during the task to engage in their naturalistic behavior, as opposed to the prior studies that have tightly controlled behaviors for neural data analysis. With this naturalistic task design, animals can perform the task without extensive training, and task performance can be comparable to humans, validating the generalizability of the task and behavior across species[27].

Second, since mental simulations likely rely on recurrent dynamics involving populations of neurons, we sought to examine the mental simulation hypothesis both at the level of single neurons and at the population level. To achieve this, we performed large-scale electrophysiological recordings from the dorsomedial frontal cortex (DMFC), which is known to play a crucial role in ongoing dynamic computations such as timing[28–35]. In addition, we capitalized on recently developed analysis tools and techniques to probe the large-scale neural data we collected[36,37]. Specifically, we employed recurrent neural network (RNN) models as an unbiased platform to test the mental-simulation hypothesis. We trained RNNs on the mental pong task with different loss functions to instantiate different model classes with or without the mental simulation capacity. By comparing the structure and dynamics of the DMFC's and RNNs' activity patterns, we assessed whether the RNNs with simulation capacity better accounted for the neural population dynamics in DMFC. This test is deemed more critical in light of recent advances showing artificial neural networks without appropriate inductive biases[4,38] can perform mental computations with dynamics that differ from those observed in biological networks[39–41].

We found that DMFC neurons encode sensorimotor information in a multiplexed manner and exhibit population dynamics that are consistent with the simulation of ball position when it is invisible. In addition, consistent with the mental simulation hypothesis, RNNs with simulation capacity generated dynamics that more closely resembled DFMC than models that were not constrained as such. Overall, these findings provide neural support for the mental simulation hypothesis in the context of physical scene understanding in a relatively naturalistic setting with free eye and hand movements.

## Results

### Task and behavior

We devised a task where subjects used a joystick to control a paddle's vertical position to intercept a ball moving across a two-dimensional frame with reflecting horizontal walls (Fig. 1A). The ball's initial position and velocity (speed and heading) were randomly sampled on every trial with the constraint that the ball reaches the endpoint within a given time window either along a straight line or after bouncing off the top or bottom wall (see "Methods"). The frame contained a large occluder covering all trajectories before the interception point. As a result, the ball was visible only during the first portion of the trial and invisible afterward. We refer to this task as mental pong because of its similarity to the computer game Pong, and because of the presence of the occluder that necessitates mental (as opposed to visually driven) computations.

Previously we reported the behavior of two monkeys trained to perform the mental pong task[27]. Here, we reiterate three relevant observations. First, the experiment involved many parametric variations of initial ball positions and velocities and included held-out conditions to test for generalization (see "Methods" for details). Animals were able to solve the task and generalized rapidly in held-out conditions without a drop in performance (Fig. 1B)[27]. This finding suggests that animals used a flexible strategy and not a fixed stimulus-response mapping to solve the task. Second, animals were free to move their eyes while manipulating the joystick. Eye movements were variable but the distance between gaze and ball positions got smaller within the trial and remained small during the occluded region (Fig. 1C and Supplementary Fig. S1). Assuming that gaze positions approximately reflect the internal estimate of attended objects[42–46], animals' eye movements are suggestive of an underlying simulation process. Note, however, that the animals' gaze did not continuously track the ball[47] as there were numerous sporadic saccades to the paddle or other intermediate points between the paddle and ball[27]. Finally, the animals were free to move the joystick/paddle throughout the trial. Similar to eye movements, joystick movements were such that the paddle got increasingly closer to the interception point but with a delayed time course compared to the eye movements. This observation suggests that animals initially tracked the ball with eye movements and used the acquired information to control the joystick afterward.

### Single-neuron response profiles

During physiology experiments, we performed large-scale recordings from neurons in the dorsomedial frontal cortex (DMFC; Fig. 2A) while monkeys performed 79 conditions of mental pong (Fig. 2B). We focused on DMFC because neurons in this area exhibit rich and heterogeneous dynamics during sensorimotor tasks[28,30–32,48,49] that have been implicated in timing[28–35] and mental simulation[14]. We recorded from a large population of individual neurons (1889 neurons; 1552 in monkey P, 337 in monkey M; Supplementary Fig. S2A, B). Initial qualitative analysis indicated that individual DMFC neurons had complex dynamics in both visible and occluded epochs with various degrees of spatiotemporal selectivity (Fig. 2C). Some neurons exhibited transient responses to the salient events such as the start of ball movement or the start of the occlusion, while others showed sustained and temporally modulated activities throughout the task (Fig. 2C and Supplementary Fig. S2C). The activity profiles for individual neurons did not have a strong relationship to the interception point (Fig. 2C; color codes in Fig. 2B). Some neurons showed spatial selectivity with respect

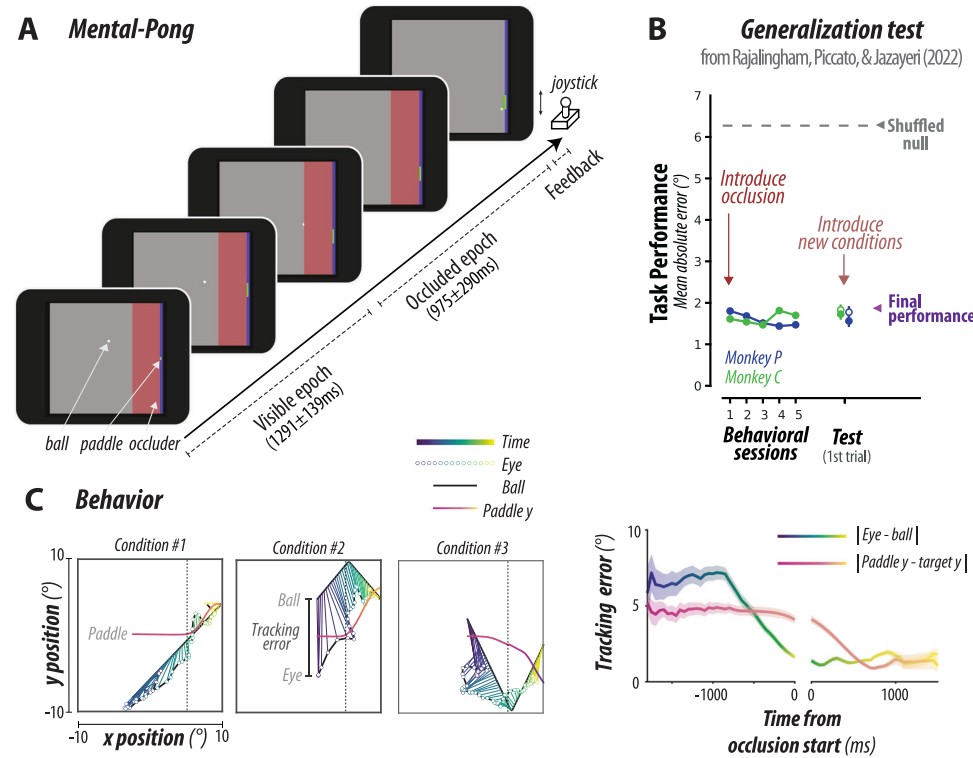

**Fig. 1 | Task and behavior. A** Mental-pong task[27]. The time course of a trial is shown. The task goal is to intercept a ball moving across a two-dimensional frame with reflecting walls, by manipulating a joystick to control a paddle vertically. The frame contained a red rectangular occluder, making the ball visible only early in the trial and, therefore, necessitating mental (as opposed to visual) computations. Monkeys freely moved their eye gaze after initially fixating on the center. The paddle was initially at the midpoint, but monkeys could drive it freely as soon as the ball moved. **B** Generalization test. Previously[27], we examined animals' performance when we introduced the occluder for the first time and when we subsequently introduced novel conditions. We measured task performance using mean absolute error between the paddle and the ball's endpoint (visual angle of degree). When compared to chance-level (gray dashed line; shuffling correct output across conditions), animals showed smaller errors from the first session with occlusion. Animals maintained similar task performance in novel conditions (closed circles, mean ± SEM across 112, 106 conditions for monkey P, C, respectively; open circles: conditions with prior exposure, mean ± SEM across 150 conditions; right arrowhead: final performance). New conditions were randomly chosen in terms of horizontal and vertical positions and velocities of the ball (in-distribution generalization). **C** Behavior. (left) Three example conditions are shown with a ball trajectory (dark blue) and eye positions (circles, color-coded by time). Lines connect the corresponding eye and ball positions in the same color code for time, with the line length denoting the eye-ball tracking error. The vertical position of the paddle (magenta line) is shown at the corresponding ball x position. The paddle was initially at the vertical center, and animals typically started moving the paddle when the ball became occluded. Eye and paddle positions are averaged across trials for each task condition. (right) Tracking error between eye and ball (blue to green) and between paddle and target (magenta to yellow) is shown as a function of time from occlusion start (line: mean, shaded region: SEM across conditions). Source data are provided as a Source Data file. Figure 1A, B adapted from Rajalingham, R., Piccato, A. & Jazayeri, M. Recurrent neural networks with explicit representation of dynamic latent variables can mimic behavioral patterns in a physical inference task. Nat. Commun. 13, 1–15 (2022) under a CC BY license: https://creativecommons.org/licenses/by/4.0/.

to the horizontal position, which is closely related to elapsed time in the trial, with a moderate dependence on the vertical position (Fig. 2C and Supplementary Fig. S2C). Overall, response profiles were characterized by strong temporal modulation and relatively weak spatial tuning, which is consistent with previous single-unit recordings in DMFC[50–54].

Given these qualitative observations, we next constructed a generalized linear model (GLM) to quantify how individual neurons encode ball position and speed (Fig. 2D). Since motor variables, including eye and joystick movements, had moderate degrees of covariation with ball position (Fig. 1C), any relationship between firing rates and ball position/speed may be explained by the motor variables. We, therefore, augmented the GLM analysis with nested models that included moment-by-moment eye and hand positions and speeds as auxiliary explanatory factors (Fig. 2D) and used it to examine the coding properties of each neuron with 10-fold cross-validation (see "Methods").

We first confirmed that the models provided a good fit for the actual firing rates for both visible and occluded epochs (Fig. 2E). Next, we partitioned the explained variance accounted for by the ball, eye,

and hand variables into common and private proportions (Fig. 2F). This approach, known as variance partitioning[55] or commonality analysis[56], is crucial for determining the contribution of each task variable when they are correlated with each other. At one extreme, all ball-related information in DMFC may be due to correlations between ball, eye, and hand variables (Fig. 1C). If so, we would expect the common variance between the ball-related variables and motor-related variables to be relatively high. At the other extreme, DMFC may carry information about ball variables independently from movement-related variables, in which case the private variance for each variable should be high. Results revealed that the ball-related variables made a strong private contribution to the explained variance in spite of correlations with other motor variables (Fig. 2F). The ball-related variables had the highest variance for the visible epoch and the second highest for the occluded epoch ($p < 10^{-10}$ and $p < 10^{-54}$ for the visible and occluded epoch, respectively; post-hoc Wilcoxon sign-rank test between the ball-related and the next highest variables). For both epochs, the ball-related variables were the most dominant in all three variable categories for unique variance. This finding suggests that DMFC carries strong information about the ball position and speed,

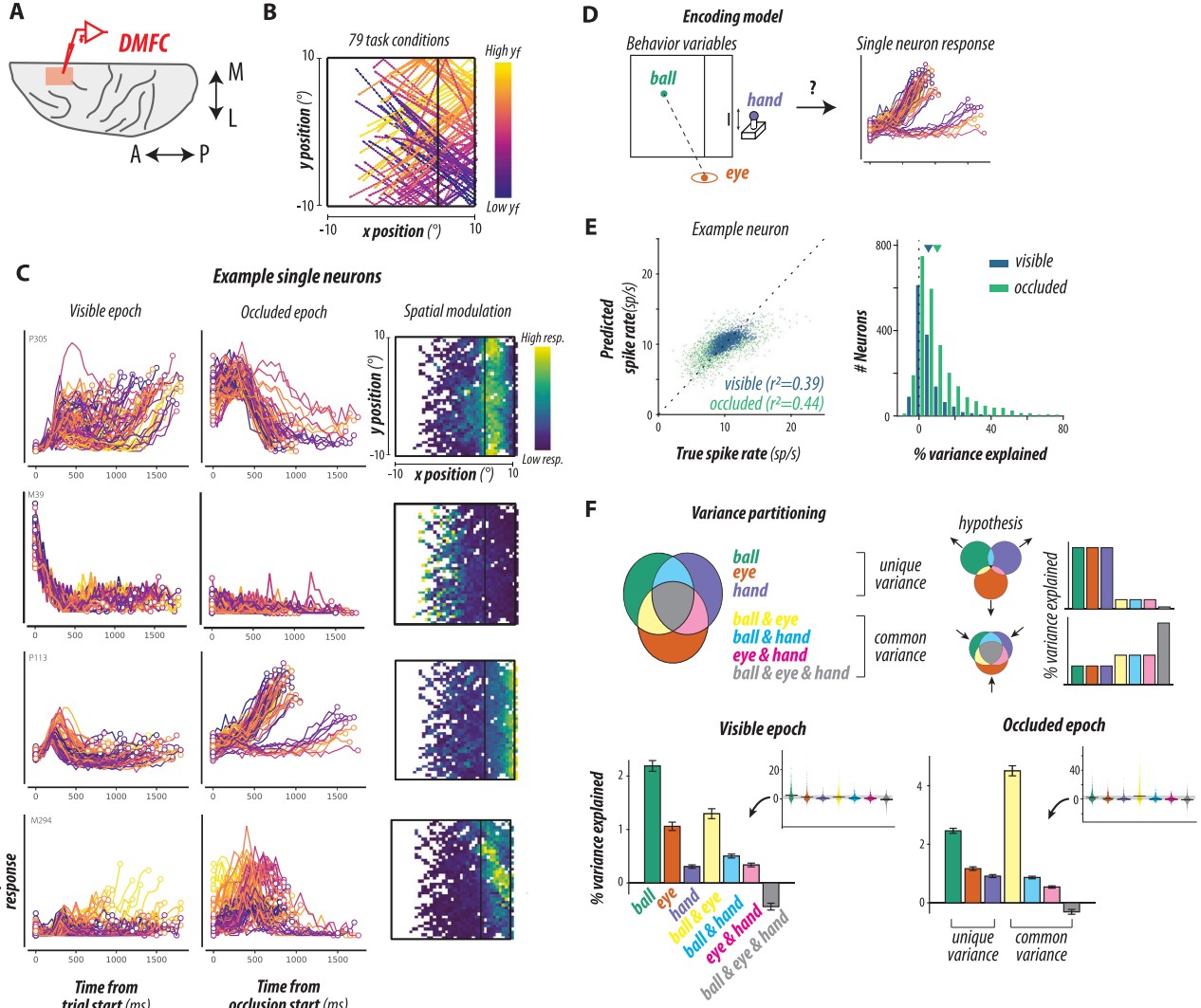

**Fig. 2 | Single-neuron responses and behavioral variable relationships.**
**A** Recording site: Neurophysiological data were collected from the dorsomedial frontal cortex (DMFC; transverse view, A: anterior, P: posterior, M: medial, L: lateral). **B** Task conditions: Ball trajectories for 79 conditions are shown; colors represent endpoint y-position (yf). **C** Example neurons: Four example neurons' activity is shown in each row. Left: Average responses across 79 conditions during the visible epoch, aligned to trial start. Middle: Responses during the occluded epoch, aligned to occlusion start. Color coding is the same as in (**B**). Right: Activity over visuospatial dimensions, with the mental pong frame's boundary in black. **D** Encoding model: We modeled how DMFC neurons encode behavioral variables for the ball, eye, and hand. For the ball and eye, we used time-varying position and speed in Cartesian or polar coordinates. For the hand, we included paddle position and joystick deflection. **E** Model fitting results: Left: Predicted and actual spike rates

for a representative neuron during visible (blue) and occluded (green) epochs. Right: Percent variance accounted for by the encoding model across neurons, with triangles marking the mean for each epoch. Negative values result from cross-validation. **F** Variance partitioning analysis: Top-left: A Venn diagram shows how neural data variance can be decomposed into unique variances for the ball, eye, and hand, as well as common variances among different combinations of the variables. Unique and common variances were parsed out by fitting nested models. Top-right: Two variance hypotheses—one with dominant unique variances, the other with more common variance overlap. Bottom: Percent variance (uncorrected for noise) for unique and common variances is shown across neurons for visible (left) and occluded (right) epochs (mean ± SEM for $N = 1385$ in visible, $N = 1389$ in occluded). Insets illustrate the distribution of variance components across neurons. Source data are provided as a Source Data file.

even after accounting for the contributions of other movement variables.

## DMFC Population dynamics

The single-neuron analyses demonstrated that DMFC neurons carry moment-by-moment information about the ball during the task. However, individual neurons had complex and heterogeneous response profiles and did not afford a coherent understanding of the ongoing computations. We, therefore, shifted our perspective away from single neurons and examined the structure of neural activity across the population (Fig. 3A). As a first step, we tested if DMFC population dynamics could be explained in terms of a small number of

latent factors, as consistently demonstrated in recent studies[28,30–32,48,49]. Results from factor analysis confirmed that the population responses were low-dimensional as several latent factors explained a large proportion of variance (Fig. 3B and Supplementary Fig. S2). Similar to single-neuron activity profiles, latent factors inferred from population activity carried rich and reliable spatiotemporal information about the ball (Fig. 3A).

According to the mental simulation hypothesis, DMFC should have a representation of the ball position across trial conditions. However, this hypothesis may be realized in many different flavors. The strongest form of this hypothesis predicts that we can decode the ball position using a fixed linear decoder during both visible and

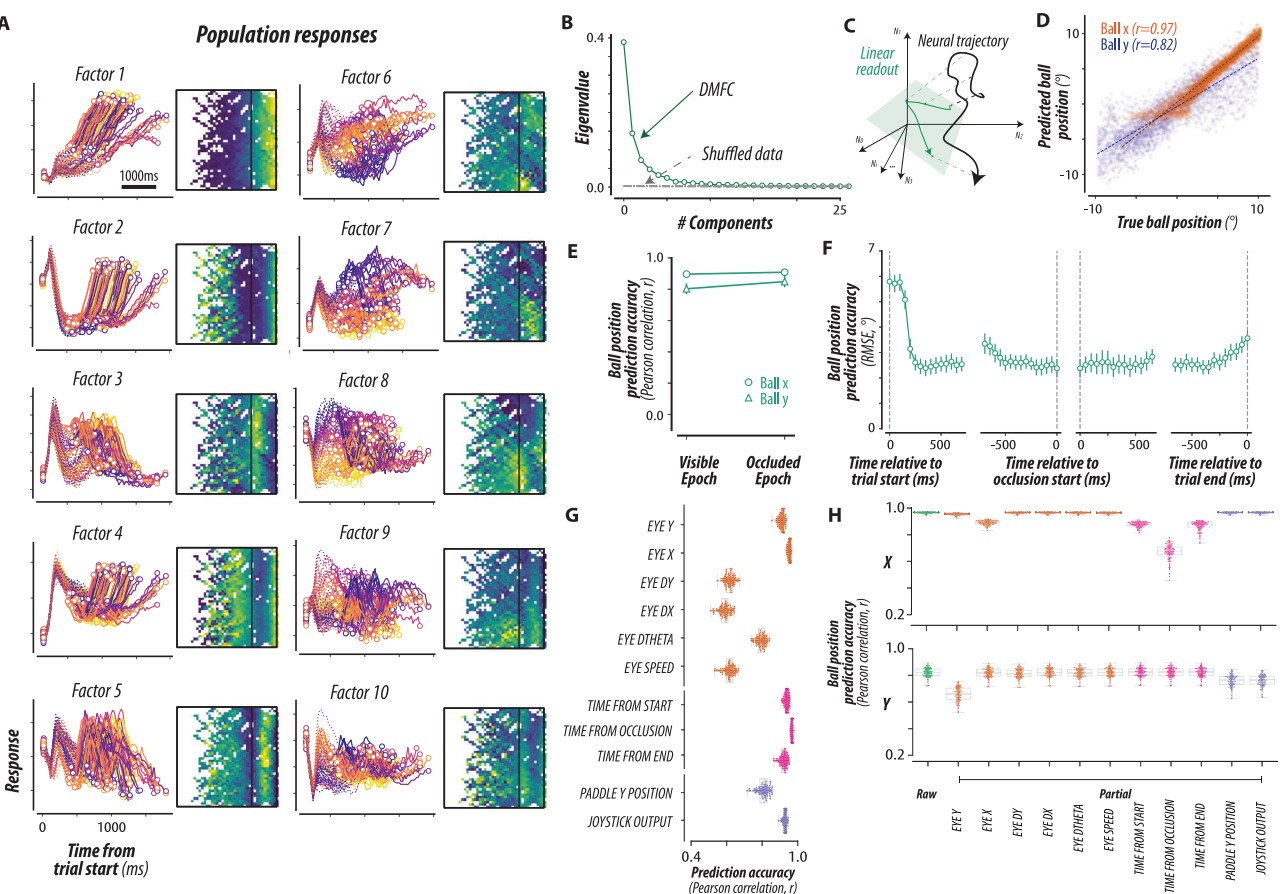

**Fig. 3 | Population responses and behavioral variable relationships.**
**A** Population responses: DMFC activity from 1889 neurons is summarized by 10 factors. Left: Time-locked responses across 79 conditions (dotted lines for visible, solid for occluded epochs, circular markers for epoch boundaries). Color coding matches Fig. 2B. Right: Heat maps of factor activity over visuospatial dimensions, similar to Fig. 2C. **B** Scree plot: DMFC neural dynamics were low-dimensional, shown by the eigenspectrum estimated using principal component analysis (dashed gray curve from shuffled data). **C** Conceptual schematic: Cross-validated linear regression was used to predict ball position from DMFC activity. Neural dynamics for one mental pong condition (black trace) were used to predict ball position (x, y) (green trace) via a linear readout (green subspace. **D** DMFC readout accuracy: Linear readout from DMFC accurately captured ball positions (Pearson correlation: $r = 0.97$, 0.83, $p < 10^{-100}$ for ball x, y, respectively; two-sided permutation test). Dotted lines show the least squares fit. **E, F** Temporal stability: Linear readout captured ball positions (x, y) across visible and occluded epochs (**E**) and

throughout the trial (**F**), measured by Pearson correlation and Root Mean Squared Error (RMSE; error bars: mean ± SEM). No significant differences were found between epochs ($p > 0.05$, two-sided permutation test; $N = 100$ splits of train and test sets). **G** Sensorimotor information: DMFC activity encoded hand and eye kinematics (position, velocity, acceleration), and trial timing. Cross-validated linear decoding showed accuracy across these variables ($N = 100$ train/test splits). Box plot bounds and center represent the first, second, and third quartiles, while whiskers represent minimum and maximum values in the data. **H** Sensorimotor variable control: Decoding accuracy for ball position (green bars) remained significant after controlling for sensorimotor variables using partial correlation. Partial correlations exceeded the chance for all covariates ($r > 0.6$, $p < 10^{-100}$; two-sided permutation test; $N = 100$ train/test splits). Box plot bounds and center represent the first, second, and third quartiles, while whiskers represent minimum and maximum values in the data. Source data are provided as a Source Data file.

occluded epochs, and across time within each epoch. We refer to this as the linearly decodable state. A weaker variant of this hypothesis is that we can decode ball position, but the weights of the decoder must be adjusted with time and across the two epochs. We refer to this weaker variant as the nonlinearly decodable state.

To test these hypotheses, we constructed linear decoders to estimate the instantaneous position of the ball [$x(t)$, $y(t)$] based on the latent factors of DMFC (Fig. 3C). The key question distinguishing between linear and nonlinear code is whether a fixed decoder can extract [$x(t)$, $y(t)$] for the entire duration of the trials and would generalize across conditions. Results supported the presence of a linearly decodable state: a single decoder accurately predicted the instantaneous ball position (Fig. 3D), robustly generalized across conditions with and without bounces (Supplementary Fig. S4A), and remained stable over time during both visible and occluded epochs (Fig. 3E, F). These results were consistent across both monkeys (Supplementary

Fig. S4C, D). These observations indicate that certain patterns of population activity in DMFC consistently encode the ball position.

During the task, monkeys typically made hand and eye movements that generally co-varied with the position of the ball (Fig. 1C and Supplementary Fig. S1). While such movements have been previously interpreted as evidence of an internal tracking strategy[43,45], it is possible that the observed neural dynamics actually reflect signals related to movement preparation and execution. Consistent with the multiplexing nature of single-neuron encoding (Fig. 2F), we found that the movement-related variables could be decoded from DMFC data, including hand and eye kinematics, as well as the time within the trial (Fig. 3G). However, even after accounting for these variables using partial correlations, we still observed significant linear decoding accuracy for the ball position (Pearson correlation coefficient, $r > 0.6$, $p < 10^{-3}$ for all bars; Fig. 3H). These results suggest that the dynamics in DMFC harbor a pattern of activity that reflects the ball's moment-by-

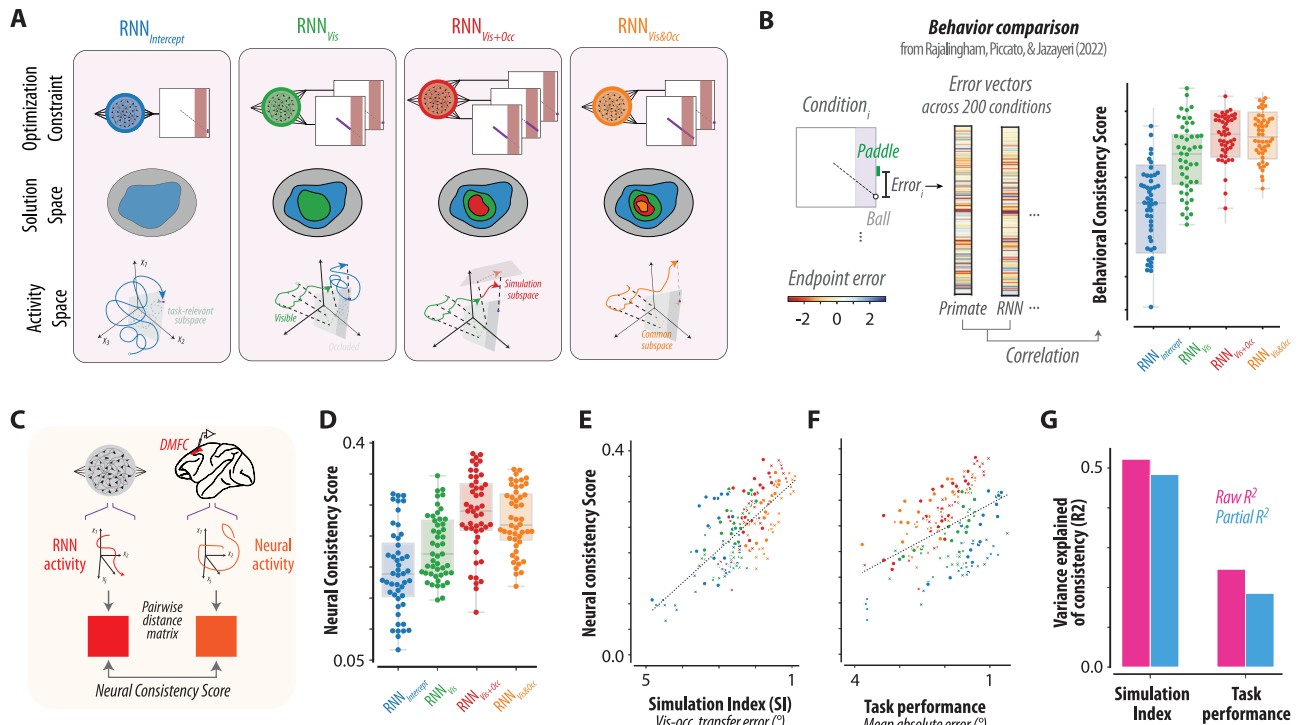

**Fig. 4 | Model-based test of simulation hypothesis. A** RNN model class: Four Recurrent Neural Networks (RNN)−RNN$_{Intercept}$, RNN$_{Vis}$, RNN$_{Vis+Occ}$, and RNN$_{Vis\&Occ}$ −were trained with different constraints to solve the mental pong task. Top: All models took visual inputs and generated time-varying outputs, linearly read out from their hidden unit activities. Optimization constraints for each RNN class were highlighted as solid in the right insets. RNN$_{Intercept}$ has one output to control paddle to intercept the ball at the endpoint. RNN$_{Vis}$ has two outputs additionally tracking the ball position during the visible epoch (solid line). RNN$_{Vis+Occ}$ has three outputs with tracking of the occluded ball position, while RNN$_{Vis\&Occ}$ tracks the ball across both epochs in one output (two outputs in total). Middle: Solution spaces (gray: entire solution). RNN$_{Intercept}$ is the least constrained (blue), while RNN$_{Vis}$, RNN$_{Vis+Occ}$, and RNN$_{Vis\&Occ}$ are progressively more constrained (green, red, and yellow). Bottom: Activity spaces. Each model's state trajectories are constrained to subspaces required for decoding the ball position during visible or occluded epochs (see "Methods"). **B** Behavior comparison: Previously[27], we compared primate and RNN behaviors across 200 conditions by measuring paddle-ball endpoint errors. RNNs trained to infer ball position during occlusion matched primate behavior

better, as reflected by higher Behavioral Consistency Scores. **C** RNN and neural dynamics comparison: Pairwise distance matrices for all states in the mental pong task were computed for both RNN and DMFC to calculate Neural Consistency Scores, representing the similarity between RNN dynamics and neural population dynamics. **D** Neural Consistency Score distribution, shown by swarm plots for each RNN, grouped by optimization type. **E** Neural Consistency vs. Simulation Index (SI). The $x$-axis is flipped to show increasing simulation capacity (left-to-right). **F** Neural Consistency vs. task performance. The $x$-axis is flipped as in (**E**). **G** SI explained more variance in neural consistency than task performance, even after accounting for covariations between SI and performance ($R^2$, pink bars; partial $R^2$, blue bars). In all panels, box plot bounds and center represent the first, second, and third quartiles, while whiskers represent minimum and maximum values in the data ($N = 48$ for each RNN class). Source data are provided as a Source Data file. Figure 4B adapted from Rajalingham, R., Piccato, A. & Jazayeri, M. Recurrent neural networks with explicit representation of dynamic latent variables can mimic behavioral patterns in a physical inference task. Nat. Commun. 13, 1–15 (2022) under a CC BY license: https://creativecommons.org/licenses/by/4.0/.

moment position that is common across conditions and generalizes between visible or occluded epochs.

An interesting question regarding the nature of ball simulation is how the ball positions are coded in DMFC during occlusion. In our single-neuron and population analyses, the ball positions were represented in the stimulus screen coordinate similar to positional variables for eye and paddle[57]. However, another possibility is that DMFC encodes the ball positions with respect to the eye positions (i.e., the egocentric framework of reference)[58]. To test this, we repeated the single-neuron encoding and population decoding analyses with egocentrically represented ball positions (Supplementary Figs. S3 and S5). Across individual neurons, our finding is mixed in that neither the egocentric nor allocentric framework dominantly explains the neurons' activity patterns (Supplementary Fig. S3). The population-level decoding also showed little difference between the two frameworks, with a tendency for better decoding in the allocentric model, particularly for the occluded epoch (Supplementary Fig. S5). These additional analyses suggest that DMFC has multiplexed information about the ball position in both coordinate systems, not clearly distinguishing between the two. This finding is also

consistent with the mixed previous reports about the coding framework in DMFC[59–61].

## Evaluating the mental simulation hypothesis using recurrent neural network models

In previous work, we tested the mental simulation hypothesis using behavioral comparisons across primates and RNNs[27]. We compared human and monkey behavior to four classes of task-optimized RNN models, each subjected to a different set of constraints (Fig. 4A). RNNs in the first class (Fig. 4A, left), which we refer to as RNN$_{Intercept}$, had only one output optimized to control the vertical position of the paddle to intercept the ball. RNNs in the second class (Fig. 4A, second from left), which we refer to as RNN$_{Vis}$, had a second output optimized to track the position of the ball in the visible epoch without any constraints on the occluded epoch (hence the subscript Vis). The addition of this output restricts the solutions to RNNs that (i) intercept the ball, and (ii) have a subspace for encoding the ball position during the visual epoch. RNNs in the third class (Fig. 4A, rightmost), which we refer to as RNN$_{Vis\&Occ}$, also had a second output that was optimized to track the ball position in both the visible and occluded epochs (hence the

subscript Vis&Occ). The addition of this output restricts the solutions to RNNs that (i) intercept the ball, and (ii) encode the ball position in the visual and occluded epochs in a common subspace. Finally, RNNs in the fourth class (Fig. 4A, third from left), which we refer to as RNN$_{Vis+Occ}$, had two additional outputs, one optimized to track the ball in the visible epoch and the other to do the same for the occluded epoch. The addition of these two outputs restricts the solutions to RNNs that (i) intercept the ball, (ii) encode ball position in the visual and occluded epochs within two subspaces that may or may not be aligned.

Comparing the behavior of these RNN classes to that of humans and monkeys, we previously reported that error patterns were best explained by RNN$_{Vis\&Occ}$ and RNN$_{Vis+Occ}$, the two model classes in which ball position during the occluded epoch is linearly decodable, as predicted by the mental simulation hypothesis (Fig. 4B). However, behavior alone cannot powerfully distinguish between model classes; even when an RNN produces behavior similar to humans and monkeys, its underlying computational strategy may differ from the brain. Here, we sought to test the mental simulation hypothesis more critically by comparing hidden state dynamics in different classes of RNN to the population dynamics in DMFC. To do so, we first computed pairwise distances between all visited states separately in DMFC and in all RNNs. This distance metric captures the structure of population activity across all mental pong conditions. We then quantified the similarity of RNNs to DMFC by measuring the noise-adjusted correlation between those pairwise distances, which we refer to as neural consistency score (see Methods, Fig. 4C). While RNNs exhibited a broad range of consistency scores within and across classes (Fig. 4D), RNN$_{Vis\&Occ}$ and RNN$_{Vis+Occ}$ showed higher overall consistency scores than RNN$_{Vis}$ and RNN$_{Intercept}$ ($p < 10^{-10}$; Wilcoxon rank sum test). This finding suggests that, among the RNN models, DMFC dynamics is more similar to RNN$_{Vis\&Occ}$ and RNN$_{Vis+Occ}$, which were augmented to have a subspace for the explicit representation of the ball position in both the visual and occluded epochs, consistent with the mental simulation hypothesis.

Next, we examined this result by comparing all RNNs on the same footing without regard to their class membership. We reasoned that comparing RNNs based on their class membership could diminish effect size because the solution space for different classes is overlapping. For instance, the solution space for RNN$_{Intercept}$ includes solutions to all other classes, and the solution space for RNN$_{Vis}$ includes solutions to RNN$_{Vis\&Occ}$ (Fig. 4A, Solution Space). To compare all RNNs on the same footing, we developed a common metric to measure the degree to which an RNN carries explicit information about the ball position while behind the occluder. The metric, which we refer to as simulation index (SI), was computed as the mean absolute error between the true time-varying ball position and the ball position predicted from a cross-validated linear regression (see "Methods"). We first validated this metric by verifying that SI was higher in RNN$_{Vis\&Occ}$ and RNN$_{Vis+Occ}$ models compared to RNN$_{Vis}$ and RNN$_{Intercept}$ ($p < 10^{-25}$; Wilcoxon rank sum test). We then examined the correlation between neural consistency score and SI across all individual RNNs. Results indicated a strong relationship between neural consistency scores and SI across the RNNs ($R^2 = 0.52$, $p < 10^{-32}$ for Pearson correlation coefficient; Fig. 4E). This result indicates that DMFC dynamics are most similar to RNNs that carry linearly decodable information about the ball position behind the occluder. The correlation between SI and consistency held for each model class with a tendency for higher correlation in RNNs with stronger SI ($R^2 = 0.51$, $p < 10^{-8}$ for RNN$_{Vis\&Occ}$; $R^2 = 0.65$, $p < 10^{-11}$ for RNN$_{Vis+Occ}$; $R^2 = 0.31$, $p < 10^{-4}$ for RNN$_{Intercept}$; $R^2 = 0.32$, $p < 10^{-4}$ for RNN$_{Vis}$; all $p$-values for Pearson correlation coefficient). Note that none of the RNNs were fit to the neural data; they were instead trained and evaluated with respect to behavior and all model parameters were reported previously before electrophysiology experiments[27]. This feature makes our comparisons between RNNs and DMFC relatively unbiased and gives more

**Table 1 | Task performance measured in terms of mean absolute error for interception point**

| Money M | Monkey P | RNN$_{Intercept}$ | RNN$_{Vis}$ | RNN$_{Vis\&Occ}$ | RNN$_{Vis+Occ}$ |
|---|---|---|---|---|---|
| 1.02 ± 0.12 | 0.84 ± 0.10 | 1.72 ± 0.14 | 1.88 ± 0.07 | 2.25 ± 0.08 | 2.02 ± 0.09 |

credibility to our model-based inference that DMFC dynamics are consistent with simulating the ball position behind the occluder.

One potential concern regarding the comparison of RNNs to DMFC is that the RNN may appear more similar to DMFC for models whose performance are closer to animals' performance. We carried out several additional analyses to address this concern. First, we compared the two animals' performance to the average performance of RNNs within each class (Table 1). We found that RNN$_{Intercept}$ and RNN$_{Vis}$ were more similar to animals' performance compared to RNN$_{Vis\&Occ}$ and RNN$_{Vis+Occ}$ ($p < 10^{-4}$; Wilcoxon rank sum test between two model categories for performance difference between RNNs and animals). This finding suggests that the reason RNN$_{Vis\&Occ}$ and RNN$_{Vis+Occ}$ were a better match for behavioral error patterns and DMFC dynamics was not task performance. Next, we considered the possibility that the correlation between SI and neural consistency scores was mediated by task performance. However, this possibility was also not supported by the data. While consistency scores were moderately correlated with task performance (Fig. 4F; $R^2 = 0.24$, $p < 10^{-12}$ for Pearson correlation coefficient), this correlation did not explain away the strong relationship between SI and consistency scores (Fig. 4G; SI: $R^2 = 0.48$, [0.42 0.56] for 95% confidence interval by bootstrapping; task performance: $R^2 = 0.19$, [0.11 0.28]). These results were consistent across both monkeys (Supplementary Fig. S6C–E). These observations suggest that DMFC dynamics carry an explicit representation of the occluded ball, consistent with the mental simulation hypothesis.

## Discrepancy between DMFC and RNN models

Our previous work comparing RNNs to behavior[27] and our current results comparing RNNs to DMFC dynamics are consistent with the idea that the brain solves mental pong via a moment-by-moment simulation of the ball position. However, this is not the only solution one can use to solve the mental pong task. A qualitatively different strategy is to predict an approximate estimate of the ball's endpoint in advance using early visual cues (i.e., ball position and speed during the visual epoch). Note that mental simulation is not inconsistent with early prediction; one can plausibly generate an early prediction using visual cues and subsequently use mental simulation to refine that estimate (Fig. 5A). We, therefore, performed further analyses to probe whether DMFC and RNNs additionally relied on an early predictive strategy.

For DMFC, we applied cross-validated linear regression to examine the relationship between the ball's endpoint and population activity early in the visual epoch (see Methods). We were able to decode the interception point accurately shortly after the ball movement onset, long before the ball reached its final position (Fig. 5B). Surprisingly when we applied the same analysis to RNN models, none exhibited this property: we did not find such rapid and sustained encoding of the endpoint from early activity (Fig. 5B). Instead, all RNNs showed a pattern of gradually increasing accuracy for the endpoint prediction. These findings highlight a clear gap between the DMFC and RNN models, demonstrating the limitations of our existing RNN models in terms of finding efficient solutions for the task and capturing neural population dynamics in DMFC.

The presence of information about the ball endpoint based on DMFC activity early in the trial is tantalizing. One possibility is that these signals reflect a behaviorally relevant process to make a rapid prediction of the ball endpoint soon after the movement onset. We cannot verify this possibility because of the inherent limitations of

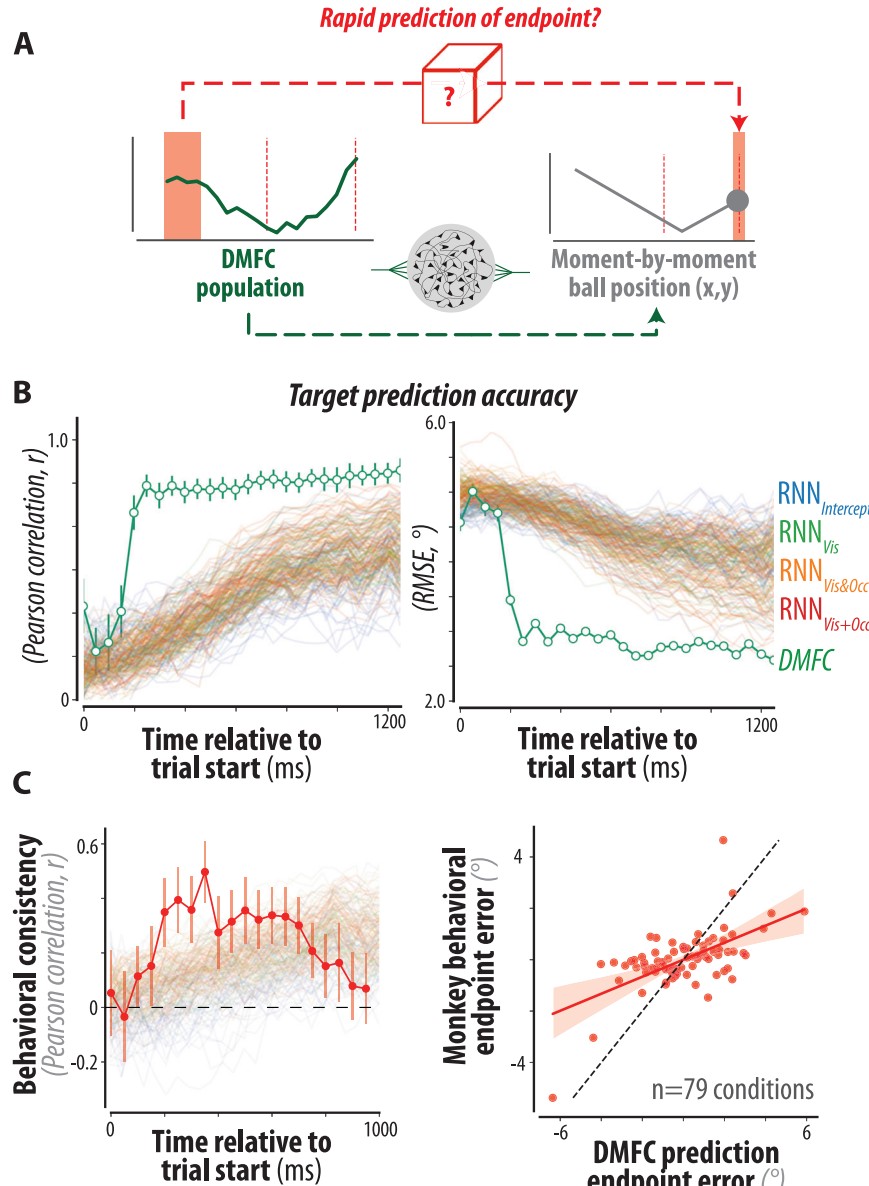

**Fig. 5 | Rapid offline prediction in DMFC and RNN models. A** Schematic contrasting the rapid offline prediction and online mental simulation processes. Rapid offline prediction (top arrow) manifests as the predictability of the ball endpoint by activity early in the visual epoch. Mental simulation (bottom arrow) manifests as the moment-by-moment representation of the ball position throughout the trial. **B** Decoding the ball endpoint in DMFC and RNNs. We quantified the accuracy of a static read-out of the endpoint ball vertical position over time, quantified using Pearson correlation (left panel) and Root-Mean-Square-Error (RMSE, right panel), for the DMFC population (green) and each of the tested RNNs (colored as in Fig. 4). Error bars denote mean ± SEM ($N = 100$ test/test splits). DMFC predictions were significantly more accurate than the shuffled control for all time points following $t = 250$ ms ($p < 10^{-10}$, two-sided permutation test). **C** Behavioral consistency of DMFC's interception point estimate. We measured the Pearson correlation between

DMFC's interception point representation and the monkeys' endpoint paddle position across conditions, after regressing out the true endpoint ball position. (left) DMFC's early interception point prediction was correlated with the monkeys' behavioral errors (red, mean ± SD across 100 random train/test splits). Behavioral consistency was significantly different from zero for all time points after $t = [250\ 400], [550\ 750]$ ms ($p < 10^{-2}$, two-sided permutation test). The mean behavioral consistency for each tested RNN model is also shown (colored as in Fig. 4A). (right) Comparison of the endpoint error (residual after regressing out the true endpoint ball position) of the DMFC interception point estimate vs. the monkeys' behavior, for all 79 conditions at $t = 400$ ms after trial start (solid line for linear regression, shaded area for 95% confidence interval by Seaborn's regplot function). Source data are provided as a Source Data file.

decoding analyses without causal manipulations. However, we performed several additional analyses to further test this possibility. First, we asked if the early prediction could be behaviorally relevant. If so, we should be able to find a relationship between this prediction and the animals' task performance across conditions. Accordingly, we computed the correlation between behavioral errors and errors in the interception point inferred from early DMFC data across conditions. Our findings showed that the neural prediction accuracy from

around 250 ms after the trial onset was predictive of the animal errors across conditions (Fig. 5C and Supplementary Fig. S7 for each monkey's data). This behavioral correlation was not observed when we repeated the decoding analysis to predict the ball's vertical position at the occlusion start, demonstrating specificity for the ball endpoint (Supplementary Fig. S8B). These observations provide tentative evidence that the predictive signals in DMFC are behaviorally relevant.

Second, we wondered if the endpoint decoding might be an artifact of our task design. We considered the possibility that the early signals, which carry information about the ball's initial position, predict the endpoint because the initial and endpoints are correlated. To evaluate this possibility, we performed a two-stage decoding. We used the initial state to decode the position and asked whether the decoded position could be used to infer the endpoint. This position-mediated decoding strategy was unable to predict the endpoint with the same speed and accuracy as the DMFC's original prediction (Supplementary Fig. S8D). Next, we performed the same two-stage analysis but considered both the position and the velocity as intermediary variables. This decoder was able to predict the endpoint accurately and rapidly (Supplementary Fig. S8D), suggesting that the original DMFC decoder might have leveraged this kinematic information to predict the endpoint. However, we observed a small difference between the two-stage and the original DMFC decoder. When we limited these analyses to conditions with no bounce where we expected the difference to diminish due to high temporal predictability, we found a similar difference in decoding performance between the kinematics-mediated decoder and the direct DMFC decoder (Supplementary Fig. S8E). In sum, we think that the small difference between the two-stage decoder based on position and velocity and the original DMFC decoder invites caution against the strong interpretation of the early prediction result. Speculations about the relevance of early prediction aside, the discrepancy between DMFC and RNNs highlights the need for additional models that would manifest dynamics commensurate with both early prediction and online simulation.

## Discussion

The ability to make predictions and inferences flexibly about the movements of objects is critical to survival and ubiquitously observed across different animal species, including non-human primates[27] and human infants[62]. A dominant cognitive theory posits that we perform these computations using mental simulations[3–5]. In this study, we aimed to test this hypothesis rigorously using a novel behavioral task, analysis of population neural activity in the frontal cortex, and modeling. We found that the structure and dynamics of neural activity in DMFC track the moment-by-moment estimate of the position of a moving ball regardless of variations in the kinematics during both visual and occluded epochs of the task, consistent with the mental simulation hypothesis.

Our study is not the first to find a neural signature of occluded moving stimuli. For example, previous physiology studies in monkeys found that single neurons in frontal[19–22] and parietal[23,24] cortices were sensitive to latent variables of the sensorimotor prediction (or working memory) tasks—such as the speed of an invisible moving object. Our results complement and extend previous work in this area in several important directions, including using a rich and more naturalistic behavioral task, inferring the geometry and dynamics of neural states from population neural activity, and comparing to concrete hypotheses instantiated by recurrent neural networks.

### Behavioral task

One notable innovation of our work is the use of the mental pong. Monkeys were able to readily learn this task and generalize their knowledge to novel conditions without additional training. These characteristics are important as they suggest that inferences we make about the brain are not the result of overtraining; instead, they reflect the natural computational capacities of the monkey brain. Moreover, unlike many previous experiments that controlled animals' movements to rule out their confounding impact on neural data[19–22,24–26] or imposed no explicit task[23], we allowed animals to freely move their eyes and hands while attempting to intercept the ball. These features enabled us to investigate the neural signals associated with mental simulation within the context of ongoing behavioral outputs without

imposing unnatural sensorimotor constraints such as eye fixation, which could impact the behavioral strategies and neural data. The value of using naturalistic tasks has long been acknowledged but these experiments have become possible only recently, thanks to technologies for large-scale recordings and sophisticated analyses. In this study, we addressed these potential confounds using various analyses, including variance partitioning developed in neuroimaging to disentangle contributions of different task parameters to fMRI activation patterns[55]. Results indicated that DMFC dynamics was not related to only one variable like eye position or hand movement. Rather, the movement parameters were encoded together with the ball position in a multiplexed fashion across neurons. This mixed selectivity is a common observation in higher cortical areas[63–67] and is thought to emerge from the need to perform multiple tasks flexibly[64,68–70]. While our approach with post hoc analysis demonstrated that the simulation-related dynamics can be dissociated from sensorimotor signals in DMFC, future experiments are needed to better tease apart their relationships, for example, by imposing eye fixation during occlusion and developing new mental tasks beyond object tracking.

### Population neural activity

Recent theoretical and experimental advances have highlighted the importance of analyzing population neural activity to gain insights about the nature of neural computations that are not easily discernible at the level of single neurons[71–73]. Several studies have analyzed the geometry and trajectory of population neural activity to shed light on the organization and dynamics of latent variables that underlie cognitive computations[28,30–32,37,39,66,74–77]. In the domain of mental simulation and object tracking, previous primate electrophysiology experiments have shown that single neurons in the frontal and parietal cortices responded differently according to the visibility state of visual stimulus and its feature parameters[19–22,24–26]. For example, it was found that parietal neurons are sensitive to inferred motion stimuli, providing a single-neuron substrate for dynamic tracking of objects[23]. Building on these works, our population encoding and decoding analyses revealed the multiplexed coding of task-relevant variables across neurons and low-dimensional embedding of the stimulus kinematics in the population dynamics. This approach enables us to test the mental simulation hypothesis more directly by examining the explicit moment-by-moment representations of the occluded objects, independent of other sensory and motor signals.

Most importantly, we found low-dimensional patterns of population activity that reflected the position of the ball stably throughout the trial. The low dimensionality of the activity across the population places important constraints on the underlying mechanisms that cannot be inferred from previous single-neuron studies[28,30–32,37,39,68,78–82]. Specifically, it suggests that either the effective connectivity in DMFC during mental pong is low-rank, or that the signals in DMFC are dominated by low-dimensional inputs[79,83]. One potential circuit-level hypothesis born out of our findings is that the other frontal[20,22] or parietal[23,24] areas provide sensory input related to the ball, thus setting up the initial conditions for the population dynamics in the DMFC that unfold during occlusion[32]. Continued advances in large-scale recording technology will afford detailed study of the within- and between-region interactions even at single-trial levels.

One intriguing result from our analysis of population neural activity was tentative evidence for rapid prediction of ball endpoint from early DMFC activity. We found that this early predictive signal was correlated with the animal's behavior. However, with decoding analysis alone, we cannot ascertain that animals actively rely on such rapid prediction. The circuit-level mechanisms that might afford such rapid prediction are also not known. One possibility is that the offline prediction depends on a qualitatively similar mechanism as the online simulation that operates at a much faster timescale. We have a few

observations consistent with this notion of flexible control of simulation time scale: the early decoding of the vertical ball positions at occlusion start (Supplementary Fig. S8C; but note that this prediction of occlusion point has no behavioral relevance, Supplementary Fig. S8B) and the strong contribution of the ball's early kinematic information to the endpoint prediction (Supplementary Fig. S8D). Such rapid dynamics may alternatively rely on different feedforward processes within the dorsal visual stream, similar to how the ventral stream is thought to support rapid object recognition[84,85]. It is also possible that the brain does not engage in any such rapid predictive process in our task. Instead, the fast prediction may be associated with the optimization of neural responses to create behavioral biases that are aligned with the prior statistics in task parameters (e.g., distribution in initial points and speeds). Further studies are needed to verify the presence of fast offline predictions and their neural mechanisms and functional relevance in behavior.

### Recurrent neural network modeling

Recurrent neural networks have helped the fields of systems and computational neuroscience[69,86,87], both for analyzing large-scale neural data[88,89] and as models for cortical dynamics[37,90–92]. In the latter context, RNNs are typically trained to perform specific tasks (i.e., generate specific time-varying outputs when the network receives sequential inputs), and their hidden activity patterns are analyzed to gain insights that are not easily obtained from neural data - for example, investigating attractor dynamics[37] or examining the effect of targeted perturbation[28]. These models were not previously used to assess the neural basis of mental simulation. In our study, we trained different families of RNN models on the mental pong task to set up an unbiased testing ground for the simulation hypothesis. Interestingly, RNN models were able to compute the endpoint through a nonlinear function of the ball's visible states without explicitly tracking the ball, demonstrating that solving mental pong does not necessitate mental simulation. Task performance was comparable across different model classes, allowing us to put all the models on equal footing except for the feature related to the mental simulation.

The essence of RNN training was to incorporate mental simulation into the networks by imposing their linear readout of hidden states to track the ball for the task. This approach of task optimization allowed us to test directly whether and how the state-space dynamics differ between the visible and occluded epochs by imposing the simulation constraint on the same or different output channels (RNN$_{Vis+Occ}$ versus RNN$_{Vis\&Occ}$). This question has remained speculative[19] or only weak evidence exists supporting the common subspace hypothesis at the single-neuron level[25]. Our results showed that both network classes exhibited a similar level of neural consistency score (Fig. 4D) and that the decoding performance remained temporally stable across epochs (Fig. 3F). These findings suggest that DMFC uses a common, not separate, subspace for the visual encoding and dynamic inference of the ball in the task. It is worth emphasizing that these RNN models were constructed prior to the collection of any of the neural data, and are pre-registered models of neural dynamics[27]. In our previous work, we demonstrated that neural network models whose dynamics are consistent with simulations are superior to other models with arbitrary dynamics in terms of capturing behavior; we here provided further evidence supporting the notion that animals may mentally simulate the ball during the task, as reflected by the similarity between DMFC and RNNs with simulation capacity in terms of neural population dynamics.

### Open questions

Our work raises important questions for future research. First, we found that RNNs that are explicitly constrained to perform simulation are better models of DMFC. However, we do not know why the primate frontal cortex adopts this solution. What are the core optimization objectives that lead to this solution? We speculate that online simulation is among a suite of computational strategies that enable neural systems to generalize flexibly, as cognitive theories[2,6] and recent advances in machine learning indicate[92]. We also do not know how signals supporting mental simulation are coordinated between different cortical and subcortical circuits. Future inquiries are needed to investigate how the dynamic interactions within and between these regions give rise to the low-dimensional dynamics in DMFC.

Second, the discrepancy between DMFC and RNNs in terms of early prediction performance of the endpoint suggests that the brain may use computations at multiple time scales – both rapid offline predictions and dynamic online tracking – to support physical inferences via interceptive movements[93,94]. This heterogeneity of solutions could explain the divergent observations regarding the need for simulation in making physical inferences and, as such, could help reconcile debates regarding the algorithms underlying our physical inference abilities[95]. We speculate that the presence of both online and offline strategies might reflect a control machinery that computes initial state estimates through a rapid feedforward computation that can be refined using online recurrent computations.

## Methods
### Subjects and surgery

Two adult monkeys (*Macaca mulatta*), one male (M) and one female (P), participated in the experiments. For each animal, surgery under general anesthesia and using a sterile surgical technique was performed to implant three pins for head restraint. A second surgery under general anesthesia and using a sterile surgical technique was performed to implant a customized rectangular recording chamber over a craniotomy targeting the dorsomedial frontal cortex in the left hemisphere from the top of the skull (Monkey P: + 24 mm posterior-anterior, 0 mm medial-lateral; Monkey M: + 24 posterior-anterior, 0 medial-lateral from interaural midpoint). The Committee of Animal Care at Massachusetts Institute of Technology approved the experiments. All procedures conformed to the guidelines of the National Institutes of Health.

### Behavioral task

In mental pong, the player controls the vertical position of a paddle along the right edge of the screen to intercept a ball as it moves rightward. On each trial, the ball starts at a random initial position $(x_O, y_O)$ and a random initial velocity $(dx_O, dy_O)$, and moves at a constant speed throughout the trial. The screen additionally contains a large rectangular occluder right before the interception point such that the ball's trajectory is visible only during the first portion of the trial. Trial conditions were constrained by the following criteria: (1) the ball always moved rightward ($dx > 0$), (2) the duration of the visible epoch was within a fixed range ([15,45] RNN timesteps or [624.9, 1874.7] ms), (3) the duration of the occluded epoch was within a fixed range ([15,45] RNN timesteps or [624.9, 1874.7] ms), (4) the number of times the ball bounced was within a fixed range ([0,1]). We previously sampled 200 conditions and reported their stimulus parameter distributions for this task in a large-scale behavioral comparison of human, monkey, and RNN across 200 unique conditions[27]. Here, we subsampled 79 conditions (26 conditions with a bounce) from this same set of conditions for neurophysiology experiments in monkeys. Stimuli and behavioral contingencies were controlled by open-source software (MWorks; http://mworks-project.org/) running on an Apple Macintosh platform.

Each trial started when the animals acquired and held gaze on a central fixation point (white circle, diameter: 0.5 degrees in visual angle) within a window of 4 degrees of visual angle for 200 ms. Following this fixation acquisition, animals were allowed to make any eye movement and freely view the screen. Afterward, the mental pong condition was rendered on the screen with the entire frame spanning 20 degrees of visual angle: the ball was rendered at its initial position

$(x_O, y_O)$, and the paddle was rendered in the central vertical position at the right edge of the frame. 300 ms after this frame was rendered, the trial began with the ball moving in its initial velocity $(dx_O, dy_O)$; this pre-trial delay was included to mitigate potential visually-driven transients in the neural recordings.

As shown in Fig. 1A, the paddle was initially rendered as a small, transparent green square (0.5 deg x 0.5 deg) but turned into a full paddle (0.5 deg x 2.5 deg) when the animals first initiated paddle movements. This feature forced animals to perform a movement (i.e., trigger the paddle) on all trials. For the remainder of the trial, animals could freely move the paddle up or down using a customized joystick. The paddle position was updated on every screen refresh (i.e., every 16.6 ms), and moved at a constant speed of 0.17 deg/16 ms = 0.01 deg/ms. A trial ended when the ball reached the right end of the screen. Upon the trial's end, the occluder disappeared to give animals feedback on their performance. If they successfully intercepted the ball, it would bounce off their paddle (see Fig. 1A), and animals received a small juice reward; if they failed to intercept it, it would continue its path off the frame. Trials were separated by an inter-trial interval of 750 ms.

## Monkey behavior
During the experiments, monkeys were seated comfortably in a primate chair, and were head restrained. For training purposes, we first acclimated monkeys to a 1-degree-of-freedom joystick placed right in front of the primate chair. Next, we started a curriculum for training animals to perform mental pong. Monkeys were first trained to use the joystick to control the vertical position of a paddle, intercepted the ball without the occluder, and then practiced mental pong using 200 unique trial conditions. We previously reported that monkeys rapidly learn to perform mental pong with the occluder on the very first session and immediately generalize to novel stimulus conditions on the first trial[27]. Specifically, we tested whether animals relied on stimulus-response memorization strategy by introducing the novel conditions randomly sampled from the 4-dimensional stimulus space (ranges of positions $x_O$, $y_O$ and velocities $dx_O$, $dy_O$: [− 8 0] deg., [−10 10] deg., [6.25 18.75] deg./sec, [−18.75 18.75] deg./sec, respectively; see Supplementary Fig. 1 of the previous paper[27] for details). The result that animals maintained comparable task performance levels suggests that this task taps into pre-existing physical inference abilities. Moreover, monkeys have remarkably human-like patterns of endpoint error over conditions, suggesting that they are a suitable model of human physical inference abilities.

For all experiments, the stimuli were presented on a fronto-parallel 23-inch (58.4 cm) monitor at a refresh rate of 60 Hz. Eye position was tracked every 1 ms with an infrared camera (Eyelink 1000; SR Research Ltd, Ontario, Canada). The joystick voltage output (0−5 V) was converted to one of three states (up: 3−5 V; down: 0−2 V; neutral: 2-3 V), which was used to update the position of the paddle. To examine how animals move their eyes and hands during trials, we first averaged the time-varying positions of the eye fixations and paddle across trials for each condition and then computed the error between eye and ball positions ('eye tracking error') and the error between the paddle and the veridical interception point ('hand tracking error') as shown in Fig. 1.

## DMFC Neurophysiology
We recorded the activity of neural populations from the dorsomedial frontal cortex (DMFC) in two monkeys while they performed mental pong. To visualize and plan the anatomical location of recording sites, we first performed a structural MRI scan where a recording grid filled with a mixture of agarose and gadolinium was inserted into the chamber. We recorded neural activity in the DMFC with 64-channel linear probes (Plexon V-probes, monkey P) and high-density silicon probes (Neuropixels, monkey M). Recording locations targeted the

region of cortex located anterior to the genu of the arcuate sulcus, and medial to the upper arm of the arcuate sulcus. We conservatively refer to this region as DMFC, potentially comprising distinct functional regions, e.g., supplementary eye field, dorsal supplementary motor area (SMA), and pre-SMA; no recordings were made in the medial bank. In monkey P, we systematically sampled DMFC by recording from all grid locations spaced 1 mm apart within an 8 mm × 3 mm region (24 distinct recording sites, each sampled in two sessions). In monkey M, we recorded from a subset of grid locations spaced 1 mm apart within an 8 mm × 1 mm region (6 distinct recording sites, each sampled in one session). To bolster the reproducibility of the neural recording process, we did not select recording locations based on the existence of task-relevant modulation of neural activity, but instead systematically sampled the cortical tissue. Recording sites are shown in Supplementary Fig. S2.

## Neural data pre-processing
We used KiloSort (KS) 3.0 to sort the spike waveform data, using all default parameters with the exception of ops.Th = [9,4]. We did not perform any post-hoc manual resorting of spike waveforms. Instead, we used a custom Python library for post-processing based on the Kilosort spike templates, which performed the removal of unstable units and merging of similar KS templates into units. This processing step acted largely as a filter: ~ 80% of output units were sorted as per KS 3.0 alone. We then visually inspected all recording sessions using summary figures for waveform similarity. Our goal was to ensure the reproducibility of the spike sorting process by using a fully automated sorting process while reserving human intervention strictly for inspection and validation.

For each recording session, we averaged the spike trains (at 1 ms resolution) across repetitions of the same task conditions, including both success and failure trials. This resulted in a pseudo-population data matrix of size $N_{units}$ x $N_{condition}$ x $N_{time}$ for each session, where $N_{condition} = 79$, and $N_{time} = 5000$ ms. Given the large number of unique conditions (a total of $79 \times 2 = 158$ conditions, over visible and occluded task settings), and given the stability of recorded units, not all neurons were recorded during all possible conditions. We excluded neurons in which more than 5 mental pong conditions were missing. For the remaining neurons, we imputed any missing values using standard matrix imputation, replacing missing entries with the global mean of the neuron's firing rate (across time and conditions). At this threshold, less than 1% of data bins were imputed.

We then pooled all sessions together, and down-sampled the trial-averaged spike responses by binning into independent time-bins of 50 ms. To avoid introducing spurious temporal dependencies, we performed no temporal smoothing of the binned spike data. The total number of recorded neurons was 2058 and 518 for monkey P and M, respectively.

We next sought to exclude units with unreliable response patterns (e.g., due to noise, low number of trials, etc.). For each unit, we randomly split all trials ($N = 1465.4 \pm 72.54$, mean ± SEM across sessions) into two equal halves and estimated the response matrix from each half, resulting in two independent estimates of the unit's response pattern of size $N_{condition}$ x $N_{time}$. We took the Pearson correlation between these two estimates as a measure of the reliability of that neural response given the amount of data collected, i.e., the split-half internal reliability. We then excluded all units where the split-half reliability was not statistically significant ($p > 0.01$, SciPy's Pearson correlation coefficient with 3502 data points for each estimate; threshold reliability: 0.0434; median ± SEM before exclusion: $0.11 \pm 0.004$). Note that this selection process does not select for neurons modulated by task variables, but only for neurons with reliable response patterns. Finally, this resulted in a data matrix of size of size $N_{units\_all}$ x $N_{condition}$ x $N_{timebins}$. The total number of reliably

recorded neurons was $N_{units\_all} = 1889$ (1552 and 337 for monkey P and M, respectively).

We found that neural responses of $N_{units\_all}$ could be captured by a smaller number of dimensions (Fig. 3B). Thus, we used Factor analysis (with $n = 50$ factors) to capture variability in neural activity that is shared across neural populations[73], resulting in a population response matrix of $N_{factors}$ x $N_{condition}$ x $N_{timebins}$. Given that the factors are not necessarily orthonormal, we additionally performed both with and without VARIMAX alignment, with no difference in decoding results.

## Encoding analyses

We examined how individual neurons' firing rates encode various task variables using generalized linear models (GLM). The task variables included six variables related to the movement of a ball, six variables related to eye movements, two variables related to hand movements, and one nuisance variable indicating whether the task condition involved the bouncing of the ball from a wall. For the ball- and eye-related variables, we considered their time-varying positions $(x, y)$ and speeds $(dx, dy)$, as well as the polar-coordinate version of their speed to accommodate different encoding schemes. The two hand-related variables were the position of a paddle on the screen and the voltage output of a joystick. In the GLM, we used trial-averaged spike counts from each neuron's response matrix ($N_{condition}$ x $N_{time}$) as the dependent variable (2040 and 1541 counts in total for the visible and occluded epoch, respectively). We separately fitted the GLM for two task epochs using the 'glmfit' function in MATLAB, assuming a normal distribution for the noise in the mean spike counts. Since DMFC neurons have a known sensorimotor latency with respect to the task variables[96], we jittered the spike counts in time ± 300 ms (12 bins with a bin size of 50 ms) and selected the temporal jitter that resulted in the best fitting outcome. To avoid overfitting, we performed 10-fold cross-validation.

One challenge in the model fitting is that the task variables about the ball, eye, and hand can be highly correlated with each other at particular time points or task conditions. This correlation could potentially affect the interpretability of the GLM coefficients. To address this issue, we constructed nested models with different combinations of variables included and estimated unique and common variances. The unique variances represent the portion of the variance explained solely by each task variable category, while the common variances account for combinations of the variables[55]. We fitted models with only one variable category, two categories, and all three categories, and computed the unique ($U_X$) and common ($C_{X,Y}$ or $C_{X,Y,Z}$) variances as follows ($R^2_X$: percent variance explained by a model with a variable $X$ alone; $R^2_{XUY}$ for the model with two variables $X$ and $Y$ included; $R^2_{XUYUZ}$ for the model with $X$, $Y$, and $Z$ included):

$$U_{ball} = R^2_{ballUeyeUhand} - R^2_{eyeUhand} \quad (1)$$

$$C_{ball,eye} = R^2_{ballUhand} + R^2_{eyeUhand} - R^2_{hand} - R^2_{ballUeyeUhand} \quad (2)$$

$$C_{ball,eye,hand} = R^2_{ballUeyeUhand} + R^2_{ball} + R^2_{eye} + R^2_{hand} \\ - R^2_{ballUeye} - R^2_{eyeUhand} - R^2_{ballUhand} \quad (3)$$

The other unique or common variances are computed in a similar manner as described above, by modifying the corresponding variable categories. It is important to note that the unique or common variances, represented by the colored parts in the Venn diagram in Fig. 2F, cannot be directly estimated from the model itself (a circle or union of circles in Fig. 2F). Raw $R^2$ values are reported without correction of noise for each neuron and each model.

## Decoding analyses

To test the capacity of neural populations to perform dynamic tracking, we used cross-validated linear regressions to read out specific task/behavioral variables (e.g., the $x$ and $y$ position of the ball over time) from the neural population data. Regressions were performed using the scikit-learn Python library. For all analyses, we focused on static read-outs, whereby a single regression was learned to map all time points of neural activity to the time-varying task variable. To do this, we reshaped the neural population data by pooling along condition and time dimensions, resulting in a matrix of size ($N_{cond}N_{timebins}$) x $N_{factors}$. However, we did not consider each row of this data as an independent observation. Instead, regressions were trained and tested on distinct mental pong conditions (using GroupShuffleSplit). We measured the goodness of regressions using several metrics: Pearson correlation between the true and predicted variable (with associated $p$ values from SciPy's pearsonr function), and the root mean squared error. For all analyses (except Supplementary Fig. S4A), cross-validated regressions were trained on 50% of the conditions and repeated over 100 train/test splits.

To account for possible covariations due to sensorimotor variables, we also computed partial Pearson correlations, the estimated correlation after regressing out co-varying attributes (candidate sensorimotor variable, e.g., eye position x) with a linear least squares regression.

In one analysis, we use DMFC's estimate of the ball position as the feature for regressing the final ball interception point. To do so, we first trained and tested linear regressions to predict the ball position from DMFC activity; this procedure was repeated over 100 train/test splits, and ball position predictions were averaged across these splits. The average ball position predictions were then used to train and test linear regressions predicting the final ball interception point.

## RNN models

To yield insight into the computations that generate these neural dynamics, we directly compared DMFC dynamics with the corresponding dynamics of recurrent neural network (RNN) models. In previous work[27], we constructed several hundreds of RNN models optimized to perform the same task as humans and monkeys. RNNs were trained to map a series of visual inputs (pixel frames) to a movement output, where the target movement output corresponded to a prediction of the particular paddle position at a particular time point in order to intercept the ball. Different RNN models varied with respect to architectural parameters (different cell types, number of cells, regularization types, input representation types), and were differently optimized (one of four different target outputs, either with or without dynamic inference). Critically, RNNs were not optimized to reproduce neural data, only to solve specific tasks.

Critically, we trained RNNs with one of four different optimization types, which we code-named as $RNN_{Intercept}$, $RNN_{Vis}$, $RNN_{Vis\&Occ}$, and $RNN_{Vis+Occ}$. For all RNN types, one of the output channels corresponded to the paddle position, which was optimized to predict only two samples per trial: one consisting of the initial central paddle position, and the second corresponding to the paddle position at the time of interception. As shown in Fig. 4, this was the only loss term for the $RNN_{Intercept}$ class. For the remaining RNNs, we additionally estimated a loss term from some of the other channels, as the mean squared error between the channel output and a target time-varying signal corresponding to the ball's position $(x,y)$ during specific trial epochs ($RNN_{Vis}$: visual epoch only; $RNN_{Vis\&Occ}$: entire trial; $RNN_{Vis+Occ}$: separate channels for visual and occluded epochs). Therefore, the state trajectories of $RNN_{Intercept}$ in the activity space (Fig. 4A, bottom) can be arbitrarily complex so long as it terminates in a desired subspace at the time of interception. The state trajectories of $RNN_{Vis}$ during the visible epoch must reside in a subspace, allowing linear decoding of the visible ball position, and can take an arbitrarily

complex path during occlusion so long as it terminates in a desired subspace at the time of interception. The state trajectories of $RNN_{Vis+Occ}$ for the visual and occluded epochs must reside in two subspaces consistent with decoding the ball position in both epochs and must terminate in a desired subspace at the time of interception. The state trajectories of $RNN_{Vis\&Occ}$ must reside in a subspace consistent with decoding the ball position throughout the trial (both visual and occluded epochs).

RNNs were trained using the TensorFlow 1.14 library using standard backpropagation and adaptive hyperparameter optimization techniques[97]; training each RNN took approximately one day on a Tesla K20 GPU. Importantly, these RNN models are pre-registered models of neural dynamics: they were constructed and tested on behavior alone before the collection of any of the neural data.

### RNN to DMFC comparisons

Consider each representation, whether neural or artificial, as a response matrix of size $N_{units}$ x ($N_{cond}N_{timebins}$), where $N_{cond}N_{timebins}$ is the total number of neural states $N_{states}$. Each representation can be characterized via a matrix of pairwise distances between all states, resulting in a matrix of size $N_{states}$ x $N_{states}$ (Fig. 4C). We estimated the similarity (termed consistency) between two representations via a noise-adjusted correlation between their corresponding pairwise distances[98].

To do so, we first randomly split all trials into two equal halves and estimated the pairwise distance matrix from each half, resulting in two independent estimates of the system's response. We took the Pearson correlation between these two estimates as a measure of the reliability of that response given the amount of data collected, i.e., the split-half internal reliability. To estimate the noise-adjusted correlation, we computed the Pearson correlation over all the independent estimates of the pairwise distance matrix from the model ($m$) and the brain ($b$), and we then divided that raw Pearson correlation by the geometric mean of the split-half internal reliability of the same pairwise distance matrix measured for each system:

$$\hat{\rho}(m, b) = \frac{\varrho(m, b)}{\sqrt{\varrho(m, m)\varrho(b, b)}} \tag{4}$$

Since all correlations in the numerator and denominator were computed using the same amount of trial data (exactly half of the trial data), we did not need to make use of any prediction formulas (e.g., extrapolation to the larger number of trials using Spearman-Brown prediction formula). This procedure was repeated 10 times with different random split-halves of trials. Our rationale for using a reliability-adjusted correlation measure for consistency was to account for variance in the pairwise distance matrices that arise from noise, i.e., variability that is not replicable by the experimental condition, and thus no model can be expected to predict. In sum, if the model ($m$) is a perfect replica of the brain ($b$), then its expected consistency score is 1.0, regardless of the finite amount of data that is collected.

This procedure was restricted to measuring distances between states during the occluded epoch, where our previous work showed that RNNs most differ. However, results are similar when considering the full trial (Supplementary Fig. S6A,B).

### Reporting summary
Further information on research design is available in the Nature Portfolio Reporting Summary linked to this article.

## Data availability
The data generated in this study have been deposited in the Zenodo database under accession code https://doi.org/10.5281/zenodo.13952210. Source data are provided in this paper.

## Code availability
The custom scripts used for data analysis in this study have been deposited in GitHub [https://github.com/jazlab/MentalPong].

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

## Acknowledgements

R.R. is supported by the Helen Hay Whitney Foundation. H.S. is supported by IBS-R015-D1. M.J. is supported by NIH (NIMH-MH122025), the Simons Foundation, and the McGovern Institute.

## Author contributions

R.R. and M.J. conceived the study. R.R. collected the data and performed main data analyses. H.S. performed single-neuron encoding analysis (Fig. 2D–F, Supplementary Figs. S3, S5, S8), behavior analysis (Fig. 1C), reconstruction of recording sites (Supplementary Fig. S2A, B), and statistical analyses in the main text. M.J. supervised the project. R.R., H.S., and M.J. wrote the manuscript.

## Competing interests

The authors declare no competing interest.
