## [Peer Review file · Nature Communications]

Dynamic tracking of objects in the macaque dorsomedial frontal cortex

Corresponding Author: Professor Mehrdad Jazayeri

This manuscript has been previously reviewed at another journal. This document only contains information relating to versions considered at Nature Communications. Mentions of the other journal have been redacted.

Version 1:

Reviewer comments:

Reviewer #1

(Remarks to the Author)

The revised paper contains a clearer explanation of the broad rationale behind the study and of the different hypotheses that are being tested. I also appreciate the new treatment of single-unit data, as well as the expanded statistical analyses. The authors have for the most part addressed my previous comments adequately.

That said, I am still unclear on the novelty of the work. As the authors acknowledge, previous studies have reported that neurons encode aspects of occluded stimuli that would be consistent with “mental simulation”. So the novelty here is to use more modern methods: a free-viewing task, large-scale recordings, RNNs, etc. There is a suggestion that the more complex task mitigates against the use of fixed sensorimotor strategies, but this was also controlled for in previous work (e.g., Assad & Maunsell (1995) had no sensorimotor mapping at all). The other conclusion is that DMFC can multiplex different kinds of time-varying signals, but it seems that this was already known as well.

Reviewer #2

(Remarks to the Author)

Review of Rajalingham et al.

Although the paper has improved and the analysis was, some of my previous concerns were not addressed satisfactorily. The main question remains how much we learn about “mental simulation” from this study. Here the mental simulation to represent the position of a moving occluded object for less than 1s. We knew from previous studies, which have now been cited, that neurons represent the position of relevant but occluded/invisible objects.

Remaining concerns:

1. In the introduction the authors claim that they will provide new insight into mental simulation, which is a much broader concept than tracking an occluded ball. I suggest that the authors make clearer in the introduction that the type of mental simulation that they are addressing was also studied in previous work. Furthermore, some of these previous studies required the monkeys to maintain fixation and are better controlled than the present one. In fact, the authors show that the eye position depends on the position of the occluded ball which suggests that they may have been aspects of eye position. The analysis of partial correlation does not really help in this regard, because it includes the entire trajectory of the ball and would pick up the retinotopic position of the ball (the position of the ball relative to the center of gaze, see below).
2. The same issue comes up in the discussion where the authors claim about mental simulation “Behavioral studies have found support for this hypothesis but direct evidence from neural activity is lacking”, which is simply ignoring the previous and better controlled studies now cited in the paper.
3. Furthermore, due to the weak behavioral constraints, such as lack of eye position control, many aspects of the design do

not allow for an easy separation of task-relevant parameters, such as e.g. retinal speed, eye speed and position of the ball in retinotopic coordinates. We know from previous work that these parameters are represented by neurons and it is likely that they influence the regression analyses where it remains difficult to appreciate if and how these parameters are accounted for, although the variance partitioning approach helps.

I'll give one example, but there are other parameters that may be equally problematic to account for. Example: the authors aim to account for eye position in a partial correlation analysis. In a perfect analysis this analysis would fully factor out eye position (which is presumably only approximated with this method) and the position of the ball relative to the center of gaze remains. This remaining signal corresponds to the position of the ball in retinotopic coordinates and many neurons will be tuned to this feature, also in frontal cortex. Hence it is not surprising that the partial correlation does not abolish the explained variance.

Hence, I was not convinced by the statement in the discussion "thanks to technologies for large-scale recordings that afford sophisticated analyses for teasing apart mental computations from confounding sensorimotor aspects of the behavior." Did the authors indeed tease apart e.g. retinal position of the ball, ball speed in retinal coordinates, eye speed, etc.? Where in the Ms can we read about this? The authors write about mixed selectivity, but if these parameters are not tested, they might show up as mixed selectivity in parameters that are included in the analysis.

4. I still do not see the added value of the comparison with recurrent neural network models (RNNs). As I stated in my first review, it remains unclear why this particular set of RNNs is chosen (the number of layers, units, type, how/ how long they were trained, etc etc) and the outcome these analyses depend on this arbitrary choice. I remain convinced that the paper improves if the RNNs are removed. Indeed, the monkey's brain has to control the eye position, the paddles, keeps track of time etc. Hence the RNNs need to represent fewer parameters and are expected to account for less variance than the brain - that they have a lower simulation index (SI) is to be expected.

- In fact, the analysis in Fig. 5 showing information about the endpoint of the ball early in the trial also reveals that these RNN models are not capturing what is going on in the brain.

- The relation between the behavioral error magnitude and the neuronal prediction can be related to variations in the difficulty of trials, e.g. due to the presence of bounces of the ball.

5. The target prediction accuracy of the neurons that is described in Fig. 5 might be tuning to the direction of the ball, which is strongly correlated with its end position. Did the authors consider this possibility?

6. The authors introduced non-trained trial types to the monkeys to examine generalization to new conditions, but it is difficult to appreciate the value of these experiments if the authors do not list the variation in speed of the ball, its initial direction, the size of before generalization and how the new trials differed. Was this interpolation or out of distribution generalization?

7. Can the authors show how they relate to the "factors" of Fig. 3, whether they showed up as "mixed selectivity" and whether they accounted for the residual decoding of the ball position in Fig. 3H, i.e. after partialization?

- One can ask the same question about the sporadic saccades, which may also be represented in the frontal cortex and related to many of the other parameters.

- The others claim that the difference in time course between eye position and joystick imply that they have "independent dynamics". I don't think that this claim is valid. They could be part of an overarching motor plan.

8. The authors state that the joystick cause the paddle to move at a constant speed. Why was it constant? Is the gradually varying speed in Fig. 1 the result of averaging across trials?

Reviewer #3

(Remarks to the Author)

The authors have thoughtfully addressed the reviewer concerns from the last round of reviews, adding additional controls and information, and adding important caveats on the interpretation of their results. I believe this work is an important contribution to the field and have no further comments.

Version 2:

Reviewer comments:

Reviewer #1

(Remarks to the Author)

I appreciate the clarifications provided by the authors in their response, and I have no additional comments other than to congratulate the authors on an interesting study.

Christopher Pack

(Remarks on code availability)
